# Preparation and Corrosion Performance of PPy/Silane Film on AZ31 Magnesium Alloy via One-Step Cyclic Voltammetry

**DOI:** 10.3390/polym13183148

**Published:** 2021-09-17

**Authors:** Chuang Peng, Nana Cao, Ziheng Qi, Yongde Yan, Ruizhi Wu, Guixiang Wang

**Affiliations:** 1College of Materials Science and Chemical Engineering, Harbin Engineering University, Harbin 150001, China; pengchuang@hrbeu.edu.cn (C.P.); caonana0912@163.com (N.C.); y5d2006@hrbeu.edu.cn (Y.Y.); rzwu@hrbeu.edu.cn (R.W.); 2Laboratory of Nuclear Chemical Engineering and Electrochemical, Yantai Research Institute and Graduate School of Harbin Engineering University, Yantai 264006, China; qizihengwork@163.com

**Keywords:** PPy/silane film, Mg alloy, cyclic voltammetry, corrosion resistance

## Abstract

PPy/silane composite film on a magnesium alloy surface was prepared by one-step cycle voltammetry. The mixed solution of methanol and water was used as the hydrolysis solvent of a γ-(2,3-glycidoxypropyl) trimethoxysilane coupling agent (KH-560). The surface morphology of the PPy/silane film, the electro-polymerization progress of KH-560 and PPy, the influence of the silane coupling agent and the corrosion behavior of the coated AZ31 Mg alloy were all investigated. The results indicated that the PPy/silane film on AZ31 Mg alloy via one-step cyclic voltammetry could provide better corrosion protection for an Mg alloy when the volume fraction of KH-560 in the hydrolysis solution was 15% and the time span of hydrolysis was 24 h with the 5.935 × 10^−10^ A cm^−2^ corrosion current density.

## 1. Introduction

Magnesium alloy is a potential alternative material in aeronautical and automotive applications despite the fact it does not possess high corrosion resistance, which is one of the main problems restricting its applications in many industries. Coating techniques are effective methods to protect Mg alloy. Many types of them have been researched, such as chemical conversion coating, anodic oxide film, laser surface treatment and organic coating [1,2,3,4]. Recently, conducting polymers (CPs) such as polyaniline (PANI) [5], polypyrrole (PPy) [6] and polythiophene (PTh) [7] have become the research hotspot in organic coatings on the Mg alloy. Among the various kinds of conductive polymers, PPy has been used to improve the corrosion resistance of metal [8,9,10]. It is one of the most suitable candidates for corrosion protection because of its relatively easy synthesis, excellent stability and low toxicity of pyrrole monomer. Bhattacharya et al. [11] prepared polypyrrole–polysulfone (PPy–Psf) composite membranes by diffusive chemical oxidative polymerization technique of pyrrole, using FeCl_3_ as oxidant. However, Cl^-^ is easy to be doped in the traditional chemical oxidation method, which will lead to the corrosion of the magnesium alloy and increase the brittleness of the powdered polypyrrole film. In contrast, the electrochemical preparation of conducting polymers can prevent this problem, so the electrochemical method is a feasible means of preparing conducting polymers on the surface of Mg alloys. Oscar et al. [12] researched the electrodeposition of PPy on carbon steel by cyclic voltammetry (CV) and chronoamperometry (CA). The growth occurs by the slow oxidation of the deposited polymer, followed by the rapid addition of monomers. The group that PPy deposited with the technique of cyclic voltammetry showed better electrochemical, morphological, and spectroscopic properties for application. Metehan et al. [13] researched the PPy film on Mg–Al alloy, which was electro-synthesized by cyclic voltammetry (CV) in bicarboxylate electrolyte. The result displayed that the corrosion resistance of the Mg–Al alloy was improved. Srinivasan et al. [14] prepared the PPy film on an AZ31 Mg alloy in an salicylic salt electrolyte using CV. They found typical cauliflower morphology with a rough surface on PPy-coated AZ31 Mg alloy.

Mei L. [15] added TiO_2_ particles to the dodecyltrimethoxysilane (DTMS) films coated onto AA2024-T3 substrates, by using the cathodically electro-assisted deposition process. The results show that applying the method can facilitate the deposition process of silane films, giving a thicker deposit and higher coverage surface along with higher roughness and hydrophobicity, and thereby improving their corrosion resistance. Xueming W. et al. [16] prepared KH560 silane film. The results indicate that the concentration of silane solution has an obvious effect on the performance of the membrane, and the performance of the membrane is better when the concentration of the hydrolysis reagent silane is approximately 10%. Franquet et al. [17] researched that an increase in the BTSE bath concentration induces the formation of a thicker but very porous layer. Indeed, the condensation reaction of Si–OH groups to form Si–O–Si links leads to the formation of a less porous layer. This step increases the barrier properties of the BTSE film and improves the corrosion protection. The Si–OR group in silane is hydrolyzed to form Si–OH, which reacts with the metal matrix to form a hydrogen bond. The specific group on the organic functional group is connected with the organic matter, thus establishing the connection between the metal matrix and the organic material. The pH environment for hydrolysis and the condensation reaction of the silane coupling agent prepared by traditional method is contradictory. Using the electrodeposition method can solve the problem. PPy film can improve the corrosion resistance of an Mg alloy, and the combination of the silane coupling agent and conductive polymer polypyrrole can overcome the brittleness of polypyrrole film. The co-electrodeposition of conductive polymers and silane coupling agents has rarely been reported, and their combination will be more and more popular among scholars and researchers.

In this paper, PPy/silane composite film was obtained in sodium salicylicum solution which included 0.15 mol/L pyrrole and various volume fractions of silane coupling agent by one-step CV. The influence of diverse volume fractions of silane coupling agent on the corrosion resistance of the Mg alloy surface was investigated.

## 2. Materials and Methods

Specimens with the dimensions 25 mm × 20 mm × 8 mm were used for the conversion film treatment, and the surface and section of matrix were polished with 320^#^, 600^#^ and 800^#^ metallographic emery papers. The polished samples were ultrasonically cleaned for 10 min after being immersed in the mixed solution of ethanol (Zhi Yuan Chemical Reagent Co. Ltd., Tianjin, China) and acetone (Zhi Yuan Chemical Reagent Co. Ltd., Tianjin, China) (1:1 of volume ratio). Afterwards, they were cleaned with distilled water and then dried for utilization.

In this experiment, the silane coupling agent-γ-(2,3-Epoxypropoxy) propyl trimethoxysilane (KH-560) (Silicon Union Chemical Co. Ltd, Nanjing, China) was utilized. The silane coupling agent needed to be prehydrolyzed before polymerization. Different volume fractions of silane coupling agent (5%, 10%, 15% and 20%) were put into a mixed solution of methanol and water, respectively, and then hydrolyzed for 24 h at room temperature. The volume ratios of KH-560, H_2_O and methanol are listed in Table 1. Electrochemical corrosion measurements were conducted in a three-electrode electrochemical cell: the Mg alloy substrate as a working electrode, the graphite rod as an auxiliary electrode, and the saturated calomel electrode (SCE) as a reference electrode.

The PPy/silane composite film was electro-synthesized on the AZ31 Mg alloy surface by CV with 7 mV/s of scanning speed, 0 V~2 V of sweep voltage range and 7 cycles of scanning cycle.

The surface morphology of PPy/silane film was examined through scanning electron microscopy (SEM; JSM-S4800, Electronics Co. Itd, Beijing, China) made in Japan electro company with 20 kV of working voltage. The thickness of coating was calculated using DIGIMIZER by taking average of 10 values. The surface function groups of the film were analyzed by ATR-IR pattern of Fourier transform infrared spectrometer (FT-IR Spectrometer; Spectrum 100, Perkin Elmer, Waltham, MA, USA). The same three-electrode system as above was applied in the electrochemical testing. Electrochemical impedance spectrum (EIS) and potentiodynamic polarization curve were tested to characterize the corrosion resistance of PPy/silane film in 3.5 wt% NaCl solution. The testing frequency of EIS varied from 10 kHz to 0.1 Hz, amplitude was 20 mV and the scanning speed of polarization was 10 mV/s.

## 3. Results and Discussion

### 3.1. The Infrared Spectrum Analysis of PPy/Silane Film

The PPy/silane film formed on Mg alloy surface were analyzed using ATR-IR spectroscopic studies and the ATR-IR spectroscopy is shown in Figure 1.

The peaks at 1455 cm^−1^ and 1581 cm^−1^ are attributed to the absorption peaks of the C=C stretching vibration of the pyrrole ring skeleton. The stretching vibration peak of N–H is detected around 3400 cm^−1^ [6,18]. The characteristic peaks of Si–OH formed after hydrolysis are found at 3200 cm^−1^~3380 cm^−1^ because some Si–OH had no condensation reaction of dehalohydrination or dehydration with other Si–OH or metal substrate during the electro-polymerization and curing process [19]. The silanol formed after the hydrolysis of KH-560 produced dehydration condensation reaction to generate the Si–O–Si which has an asymmetric vibration peak at 1030 cm^−1^ and 1083 cm^−1^ [20]. The sharp peak at 1248 cm^−1^ is the feature peak of the epoxy group (CH_2_OCH-) [21], which illustrates that the epoxy ring was not open during the reaction. The absorption peak at 865 cm^−1^ is Si–O–Mg which was formed through connecting silanol formed after hydrolysis and the -OH on the Mg alloy surface by covalent bonds. The in-plane bending vibration peak of C–H and the stretching vibration absorption peak of C=O are detected at 600 cm^−1^~756 cm^−1^ and 1700 cm^−1^, respectively.

### 3.2. Formation Mechanism of PPy/Silane Composite Film on Mg Alloy Surface

The chemical and electrochemical reactions occurred on the AZ31 Mg alloy surface in the absence and presence of an electric field, respectively, and the formation mechanism of the PPy/silane film by one-step CV was analyzed. The surface morphology after chemical or electrochemical reaction in sodium salicylate solution is shown in Figure 2.

The surface morphology of the Mg alloys without and with an electric field are shown in Figure 2a,b, respectively. Many cracks and holes are clearly observed in the surface without an electric field, while the film after electro-chemical reaction present a uniform surface with smaller pores and no cracks. Figure 2c,d are the morphology of cross sections after the chemical and electrochemical reactions, respectively. In Figure 2d, the conversion film on Mg alloy was clearly observed, and the thickness value of the film is 28 μm. In contrast, there is clearly no layer on the Mg alloy surface according to the cross section in Figure 2c. Figure 2e shows that N elements which could be related to PPy are uniform distributed on the surface of the Mg alloys. These phenomena illustrate that the Mg alloy in sodium salicylate solution was corroded in the absence of an electric field and formed a thin conversion film in the existence of an electric field [22].

Figure 3 shows the surface morphology of an Mg alloy with chemical and electrochemical reaction for 66 min in the mixed solution containing 0.5 mol/L sodium salicylate and 15% volume fraction of silane coupling agent after prehydrolyzing for 24 h.

The thin films on the Mg alloy surface in the absence and existence of an electric field are shown in Figure 3. The thickness value of the film in the existence of an electric field (Figure 3d) was 16 μm thicker than that in the absence of an electric field (Figure 3c), which illustrates that the presence of an electric field had influence on the composite film in the mixed solution of sodium salicylate and the silane coupling agent [23]. The Si-OH through the hydrolysis of the silane coupling agent then carried on with the condensation reaction with -OH on the Mg alloy surface and connected with each other by covalent bonds in lack of an electric field. [24]. Nevertheless, with the application of an electric field, sodium salicylate formed a thin film on the Mg alloy surface and some of the silane coupling agent was chemically absorbed on the metal surface, other was electrodeposited on the Mg alloy surface. The reason is that applying an electric field resolved the contradictoriness of the hydrolysis and condensation of the silane coupling agent, and the alkaline of the Mg alloy surface was beneficial for condensation polymerization, while the rest of electrified solution was acid which was advantageous for forming Si–OH. Therefore, the solution in the presence of an electric field was beneficial for forming a silane film which was thicker than that in the absence of an electric field.

Figure 3e,f show the EDS result of the PPy/silane film. The presence of the N and Si elements is evident, as these originated from PPy and silane, respectively. The EDS result showed that the PPy/silane film suggests a uniform dispersion on the surface of Mg alloys [25].

From the above, the sodium salicylate and silane coupling agent deposited a composite film on the surface of the Mg alloy under the electric field. Thereby, when the PPy/silane composite film was formed on the Mg alloy surface by electrochemical methods, sodium salicylate formed a passive film to reduce the anodic dissolution of the Mg alloy, as the PPy film could be easily deposited by electro-polymerization. During the polymerization progress of PPy/silane film, some of the silane coupling agents connected with metal via chemical absorption on the surface of Mg alloy and others depended on electro-deposition [26]. Eventually, the pyrrole monomer and silane coupling agent were electro-deposited together on the Mg alloy surface to prepare the PPy/silane film.

### 3.3. Composite Film Prepared by Cyclic Voltammetry

With an electric field, the pyrrole monomer was electro-polymerized on the Mg alloy surface and KH-560 produced the electro-deposition and chemical absorption. PPy/silane films prepared by CV at different volume fractions of KH-560 are shown in Figure 4. The illustration in the upper-right corner in Figure 4 is the first of seven cycles of the cyclic voltammetry curves. At the beginning of applying current, the current density decreased sharply, which accounted for the reduction in the reaction rate of the Mg alloy anodic dissolution [27]. It has been proposed that SS is the most suitable candidate for the formation of adherent and homogenous PPy films on oxidizable metals by electrochemical polymerization [28]. The oxidation potential of sodium salicylate is reported to be 0.8 V. The oxidation peaks at 0.8 V in the first cyclic of Figure 4a,b indicate that the passive film was formed due to the reaction of sodium salicylate on the Mg alloy. With the growth of the volume fraction of KH-560 in the electrolyte, the oxidation peak of the passive film moved to the negative potential. When the volume fraction of KH-560 was 15%, the oxidation potential to form the passive film was 0.06 V and when the volume fraction of KH-560 was raised to 20%, the oxidation potential rose to 0.4 V. Consequently, the addition of KH-560 firstly influenced the oxidation peaks which formed the passive film of sodium salicylate on the Mg alloy surface.

In Figure 4a,b, the oxidation peaks also emerged at 0.8 V in the second cyclic, which illustrated that while sodium salicylate was continuing to form the oxidation film, pyrrole monomer began participating in the process of polymerization. With the cycle of reciprocation, the potential of the oxidation peak moved to the positive direction [29]. As more KH-560 was added in the electrolyte, the variation of oxidation potentials in Figure 4c,d are less obvious than that in Figure 4a,b. At the end of anodization scanning, the current density ascended slightly, which illustrates how the pyrrole monomer formed the active free radical cations firstly in the presence of an electric field and then formed a dimer with another active free radical cation due to its instability. With the occurrence of polymerization, the tripolymer was achieved and then the PPy film on the surface of Mg alloy was prepared. In this experiment, the silane coupling agent promoted the process of polymerization and formed the PPy/silane composite film.

### 3.4. The Influence of Volume Fraction of Silane Coupling Agent on the Morphology of Composite Film

Figure 5 shows the surface morphology of the PPy/silane film prepared at different KH-560 volume fractions (5%, 10%, 15% and 20%). As a whole, the morphology was wrinkled. As shown in Figure 5a, a great number of cracks emerged from the morphology of the film prepared with 5% silane coupling agent. With the improvement in the content of silane coupling agent to 10%, no obvious improvement of the cracks was observed. When the volume fraction of the silane coupling agent was 15%, the cracks vanished and the surface became smooth, as shown in Figure 5c. Although the addition of silane coupling agent improved the detection of the brittle crisp and bad mechanical property of the PPy film, too much silane coupling agent in the electrolyte also had negative effects on the PPy film. In Figure 5d, when the volume fraction of the silane coupling agent was increased to 20%, cracks appeared on the surface of the PPy/silane film again. The reason is that the high content of silane coupling agent decreased the stability of the solution so that the expected objective of ameliorating PPy with silane was not achieved.

### 3.5. The Electro-Chemical Influence of Silane Coupling Agent on Composite Film

The EIS results of samples containing different volume fractions of silane coupling agent are shown in Figure 6. The Nyquist plot (Figure 6a) shows that the capacitive arc radius of the 5% treated sample has the minimum diameter, and the 15% treated sample has the maximum diameter. The corrosion resistance is proportional to the diameter of the capacitive arc [30], and this indicates that the volume fraction of silane at 15% has the best anti-corrosion effect. Compared to the other samples, the volume fraction of silane at 15% has the bigger capacitive loop diameter, which is attributed to the dense surface being resistant to the corrosive ions reaching the substrate through passivation film.

Moreover, a bigger |Z| modulus tending towards 0 means better corrosive resistance, and the double logarithmic impedance versus frequency plot (Figure 6b) shows that the |Z| modulus is 5.05 × 10^4^ Ω·cm^−2^, 1.5 × 10^5^ Ω·cm^−2^, 2.69 × 10^7^ Ω·cm^−2^, and 2.37 × 10^7^ Ω·cm^−2^ for the volume fraction of silane is 5%, 10%, 15% and 20%, respectively. Compared with the |Z| modulus of the 5% sample, the |Z| modulus of the 15% sample and 20% sample increased by three orders of magnitude, and the 15% sample is highest among all the samples. The result means that the corrosion resistance increased the most when the volume fraction of silane was 15%. In addition, there are three time constants in Figure 6c, which at high frequencies (10^3^–10^4^ Hz) related to the PPy/silane film, the capacitive loop grows larger and wider, demonstrating that the PPy/silane film is dense and homogeneous when the electrolyte with 15% volume fraction of silane. The medium frequencies (10–10^3^ Hz) related to the thin film formed by sodium salicylate. The low frequency (10^−1^–10 Hz) related to the Mg(OH)_2_ film [31].

The ZSiDemo was used to fit the EIS result; the equivalent electric circuit (EEC) is shown in Figure 6d, and the electrochemical parameters of EEC are listed in Table 2. The first component indicates the resistance of solution, the next component indicates the constant phase element (CPE_1_) and the resistance of pores on the surface (R_pore_), the last component indicates the constant phase element (CPE_2_), the resistance of charge transfer (R_ct_), and the trend of diffusion (Z_w_). The values of R_pore_ and Rct increased from 2.59 × 10^3^ Ω·cm^−2^ to 2.46 × 106 Ω·cm^−2^ and from 9.70 × 103 Ω·cm^−2^ to 2.87 × 106 Ω·cm^−2^ when the volume fraction of silane coupling agent is 15%. The result indicates that increases in the values of R_pore_ and R_ct_ are related to the improvement of surface morphology. As a result, the optimal volume fraction of the silane coupling agent in electrolyte to prepare the film was 15%.

The polarization curves of PPy/silane film in 3.5 wt% NaCl solution are shown in Figure 7. Table 3 is the corresponding electrochemical parameters. With the increase in the volume fraction of silane coupling agent in electrolyte, the corrosion current density firstly decreased from 1.833 × 10^−7^ A·cm^−2^ to 5.935 × 10^−10^ A·cm^−2^ and then increased to 1.888 × 10^−9^ A·cm^−2^. The addition of silane coupling agent effectively remedied the disadvantages of the electrolyte and reduced the cracks on the film surface, which prevented corrosive solution from penetrating into film through the holes and cracks. When the volume fraction of silane coupling agent was 15%, the corrosion current density was 5.935 × 10^−10^ A·cm^−2^, which was three orders of magnitude lower than the PPy/silane film prepared at 5% and 10% silane coupling agent, and one order of magnitude lower than that prepared at 20% silane coupling agent.

The polarization curves of the PPy/silane film in 3.5 wt% NaCl solution are shown in Figure 7. Table 3 is the corresponding electrochemical parameters, and a saturated calomel electrode was used to record the potential values. With the increase in the volume fraction of the silane coupling agent in electrolyte, the corrosion current density firstly decreased from 1.833 × 10^−7^ A cm^−2^ to 5.935 × 10^−10^ A cm^−2^ and then increased to 1.888 × 10^−9^ A cm^−2^. The addition of silane coupling agent remedied the disadvantages of the electrolyte effectively and reduced cracks on the film surface, which prevented the corrosive solution penetrating the film through holes and cracks. When the volume fraction of silane coupling agent was 15%, the corrosion current density was 5.935 × 10^−10^ A cm^−2^, which was three orders of magnitude lower than the PPy/silane film prepared at 5% and 10% silane coupling agent, and one order of magnitude lower than that prepared at 20% silane coupling agent.

## 4. Conclusions

(1) The PPy/silane film without defect was successfully prepared, and the thickness value of film is 16 μm thicker in the presence of an electric field. The optimal volume fraction of the silane coupling agent in electrolyte to prepare the PPy/silane film was 15%. The corrosion current density of the PPy/silane composite film on the Mg alloy surface reached 5.935 × 10^−10^ A·cm^−2^ when the volume fraction of the silane coupling agent was 15% and with hydrolysis at room temperature for 24 h.

(2) The oxidation potential of the passive film which formed on the Mg alloy surface was impacted by the addition of silane coupling agent. The electrochemical preparation of the PPy/silane film belongs to the simultaneous process of polymerization and deposition.

## Figures and Tables

**Figure 1 polymers-13-03148-f001:**
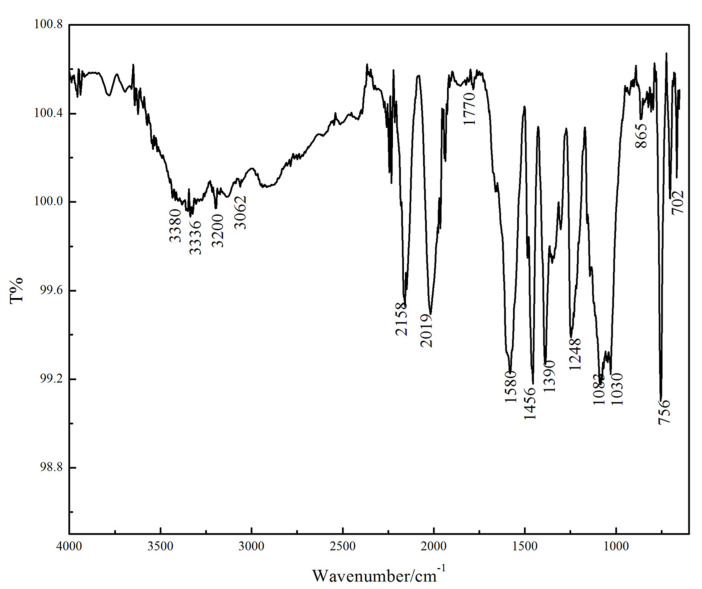
The ATR-IR spectroscopy of the PPy/silane film.

**Figure 2 polymers-13-03148-f002:**
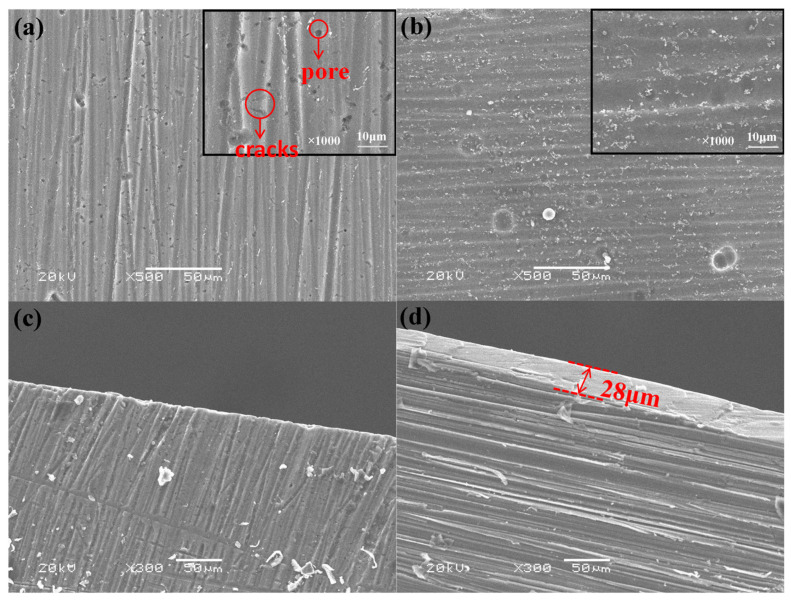
The surface and cross section morphology after chemical or electrochemical reaction in sodium salicylate solution: (**a**) the surface morphology without an electric field; (**b**) the surface morphology with an electric field; (**c**) the section morphology without an electric field; (**d**) the section morphology with an electric field; and (**e**) EDS map analysis of the element N (when with an electric field).

**Figure 3 polymers-13-03148-f003:**
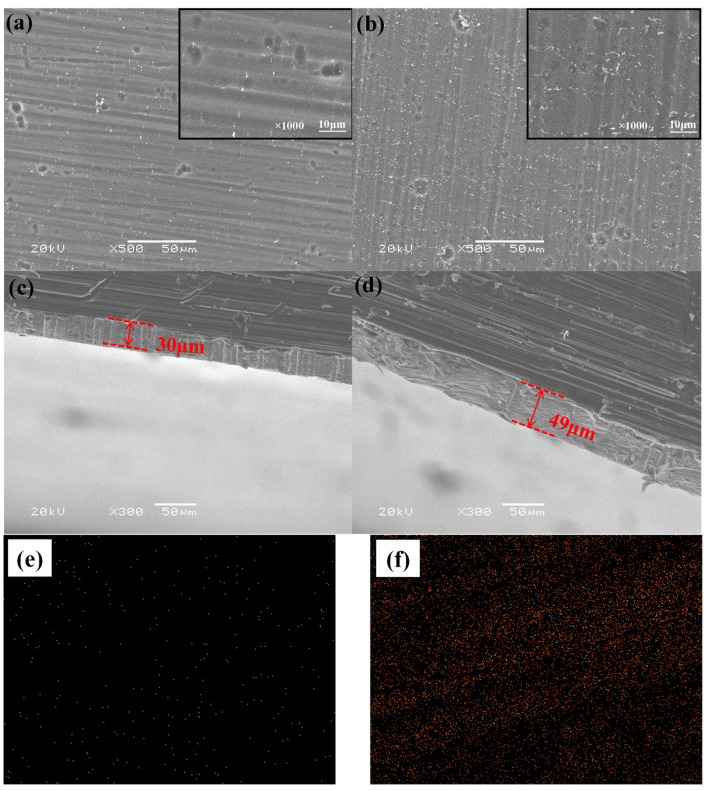
The surface and cross section morphology of the chemical and electrochemical reactions in the mix solution: (**a**) the surface morphology without an electric field; (**b**) the surface morphology with an electric field; (**c**) the section morphology without an electric field; (**d**) the section morphology with an electric field; (**e**) EDS map analysis of the element N (when with an electric field); and (**f**) EDS map analysis of the element Si (when with an electric field).

**Figure 4 polymers-13-03148-f004:**
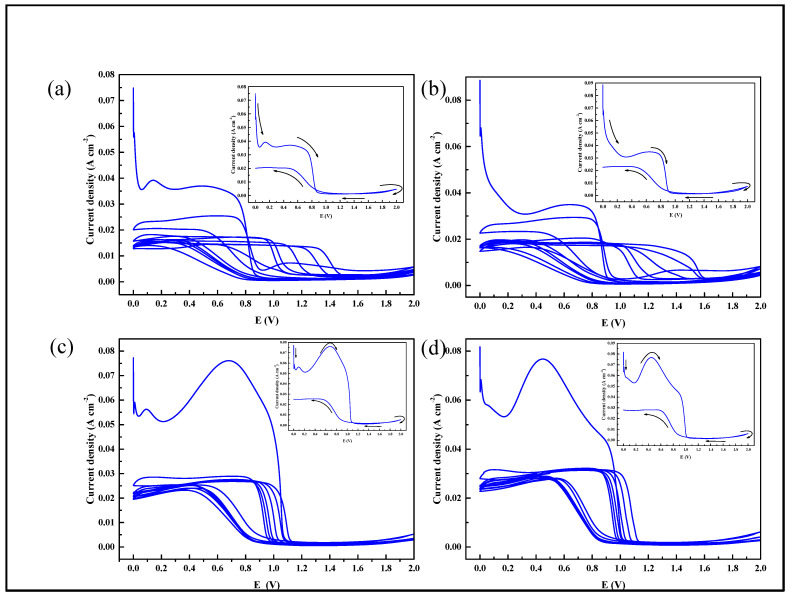
CV and the first cycle of different volume fractions of KH-560-prepared PPy/silane film on the AZ31 Mg alloy: (**a**) 5% KH-560; (**b**) 10% KH-560; (**c**) 15% KH-560; and (**d**) 20% KH-560.

**Figure 5 polymers-13-03148-f005:**
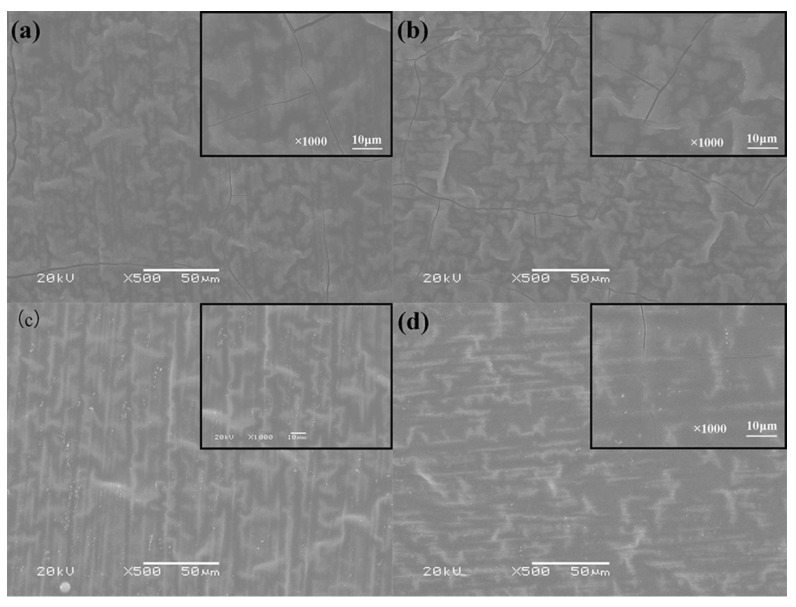
The surface morphology of composite coating compared with different volume fractions of KH-560: (**a**) 5% KH-560; (**b**) 10% KH-560; (**c**) 15% KH-560; and (**d**) 20% KH-560.

**Figure 6 polymers-13-03148-f006:**
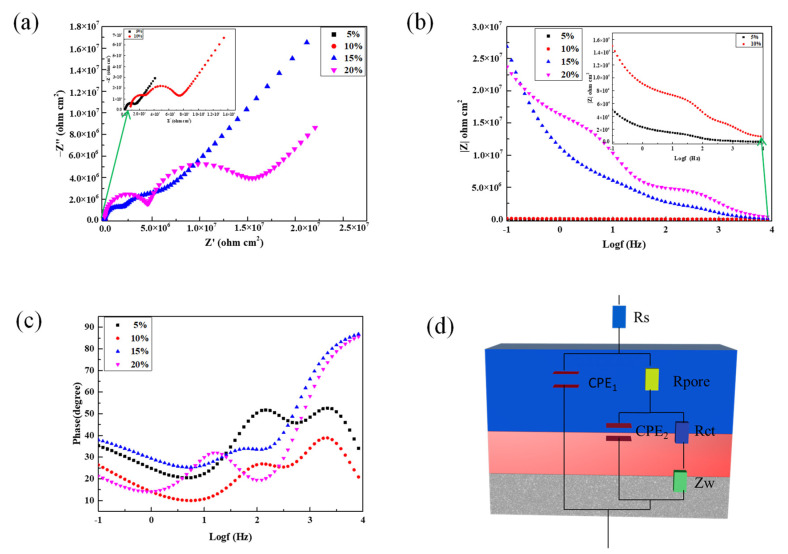
EIS of PPy/silane film prepared at different concentrations of silane coupling agent in 3.5 wt% NaCl solution: (**a**) Nyquist plot; (**b**,**c**) bode plots; and (**d**) equivalent electric circuit.

**Figure 7 polymers-13-03148-f007:**
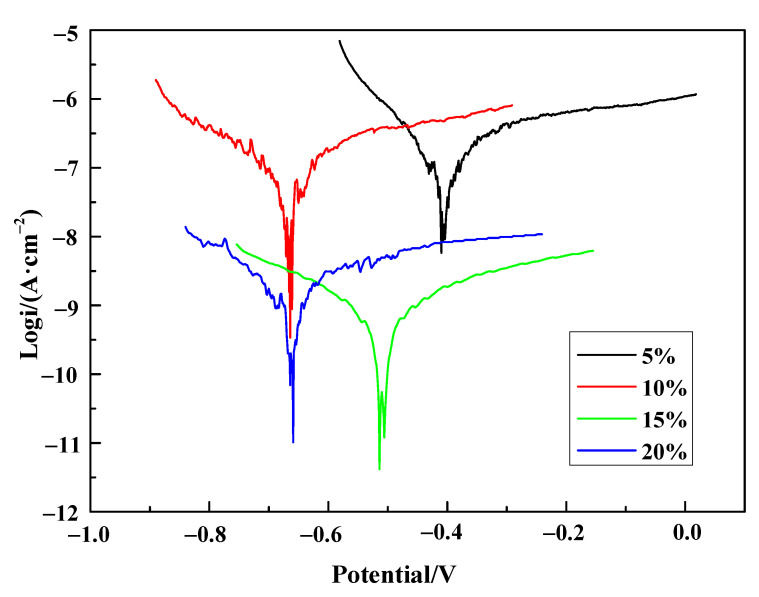
The polarization curves of PPy/silane composite at different concentrations of silane coupling agent in 3.5 wt% NaCl solution.

**Table 1 polymers-13-03148-t001:** The volume ratio of KH-560, H_2_O and methyl alcohol.

**Volume Fraction**	5%	10%	15%	20%
**Ratio**	1:18:1	1:8:1	3:14:3	1:3:1

**Table 2 polymers-13-03148-t002:** The electrochemical parameters of EEC for the different concentration of the silane coupling agent.

Concentration	R_s_(Ω·cm^−2^)	CPE_1_(F·cm^−2^)	Rpore (Ω·cm^−2^)	CPE_2_(F·cm^−2^)	R_ct_(Ω·cm^−2^)	Z_w_(Ω^−0.5^·cm^−2^·S^−1^)
5 wt%	340	7.36 × 10^−8^	2.59 × 10^3^	2.35 × 10^−7^	9.70 × 10^3^	3.07 × 10^−5^
10 wt%	790	6.02 × 10^−9^	2.66 × 10^4^	5.89 × 10^−8^	3.41 × 10^3^	1.35 × 10^−5^
15 wt%	0.0138	1.41 × 10^−10^	2.46 × 16^3^	1.26 × 10^−9^	2.87 × 10^6^	5.43 × 10^−7^
20 wt%	0.02442	5.14 × 10^−11^	9.11 × 10^5^	1.69 × 10^−9^	9.11 × 10^5^	1.07 × 10^−7^

**Table 3 polymers-13-03148-t003:** The corresponding electrochemical parameters of polarization curves.

**Concentration**	5%	10%	15%	20%
**Corrosion Current Density ((A·cm^−2^))**	1.833 × 10^−7^	1.736 × 10^−7^	5.935 × 10^−10^	1.888 × 10^−9^
**Potential (V)**	−0.410	−0.664	−0.576	−0.659

## Data Availability

Not applicable.

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
