# Peer review of "Preparation and Corrosion Performance of PPy/Silane Film on AZ31 Magnesium Alloy via One-Step Cyclic Voltammetry"

_polymers, 2021, doi:10.3390/polym13183148_

Round 1
Reviewer 1 Report
The authors of the paper: Preparation and corrosion performance of PPy/Silane Film on AZ31 Magnesium Alloy via One-Step Cyclic Voltammetry present some interesting experimental results obtained from a new deposition technique on metallic substrate. Some clarification are necessary to improve the paper:
L23: for what kind of applications AZ31 alloy represent an alternative ?
L108: improve the quality of figure 1, especially axes
L112 : 6,8 references on the same [6,8]
L132 and 133: remove one of the words morphology
L140: indicate the film in figure 2d , it looks like the other face of the substrate in a different angle view . Confirm the presence of the polymer layer on transversal images from figures 2 and 3 with EDS determination or other technique
Line 228: highlight the differences between figure a) and b) - at the first look there are no differences , in figure 5 b) the cracks are also present as in a) with similar dimensions
Reviewer 2 Report
The article presents the production method and investigates the surface and cross-section morphology as well as the corrosion resistance of PPy/Silan composite layers on the AZ31 alloy surface. The presented research results are of a utilitarian nature in the anti-corrosion protection of magnesium alloys.
Notes to the article:
Lines 54-55: „Wang et al. [16] prepared KH560 silane film”. See References [16] on lines 341-342: „Xueming, W.; Aiju, L.; Guoli, L.; Weiqiang, W.; Yonghui, L.; Studies of the Preparation …………….”.
Also check "Li et al. [15]…. " on line 50.
Figure 1: The descriptions of the axes in the chart are unreadable.
Figure 2: The descriptions in the figure are not very contrasting.
Line 134-136: The authors say that cracks can be seen in Figure 2a, b. It is not known where these cracks are visible. In my opinion, these are surface irregularities, not cracks.
Figure 2: The article does not describe how the samples were prepared for the study of the structure on the cross-section. Were the samples mounted, grinded and polished? There appear to be deep parallel scratches on the surface which may be a cut or polishing residue. This may affect the quality of the results obtained.
Line 140: On what basis the minimum and maximum coating thickness was determined. There are also no measurement errors.
Figure 3: The descriptions in the figure are not very contrasting.
Lines 158-169: The described mechanism and reasons for the reactions taking place require evidence or reference to the literature. Moreover, it is highly unscientific to claim that the resulting layer is "thicker" or "thinner" only on the basis of a visual assessment of the cross-section. It is not understood why the thickness of the coatings was not measured. I believe that these measurements need to be completed.
Figure 4. Charts and descriptions are illegible.
Figure 6: The descriptions in the figures are hardly legible.
Figure 7, Table3: Specify the electrodes against which the potential values were recorded. In addition, Figure 7 and Table 3 should be placed after the paragraph on lines 271-281.
In Conclusions, it should be also reference to the structure and thickness of the obtained layers.
Round 2
Reviewer 1 Report
Publish in this form
Reviewer 2 Report
The authors significantly improved the article. They have addressed all my comments and incorporated them into the text. They supplemented the comments, references to literature and corrected the drawings.
I believe that the article may be published.